# Measles Virus-Induced Host Immunity and Mechanisms of Viral Evasion

**DOI:** 10.3390/v14122641

**Published:** 2022-11-26

**Authors:** Lucia Amurri, Olivier Reynard, Denis Gerlier, Branka Horvat, Mathieu Iampietro

**Affiliations:** 1Centre International de Recherche en Infectiologie (CIRI), Team Immunobiology of Viral infections, Univ Lyon, Inserm, U1111, CNRS, UMR5308, Université Claude Bernard Lyon 1, Ecole Normale Supérieure de Lyon, 21 Avenue Tony Garnier, 69007 Lyon, France; 2Centre International de Recherche en Infectiologie (CIRI), Team Neuro-Invasion, TROpism and VIRal Encephalitis, Univ Lyon, Inserm, U1111, CNRS, UMR5308, Université Claude Bernard Lyon 1, Ecole Normale Supérieure de Lyon, 69007 Lyon, France

**Keywords:** Measles virus, innate immunity, immune evasion, signaling, immune amnesia, sensors, interferon, cGAS/STING, RNA virus, DNA sensing

## Abstract

The immune system deploys a complex network of cells and signaling pathways to protect host integrity against exogenous threats, including measles virus (MeV). However, throughout its evolutionary path, MeV developed various mechanisms to disrupt and evade immune responses. Despite an available vaccine, MeV remains an important re-emerging pathogen with a continuous increase in prevalence worldwide during the last decade. Considerable knowledge has been accumulated regarding MeV interactions with the innate immune system through two antagonistic aspects: recognition of the virus by cellular sensors and viral ability to inhibit the induction of the interferon cascade. Indeed, while the host could use several innate adaptors to sense MeV infection, the virus is adapted to unsettle defenses by obstructing host cell signaling pathways. Recent works have highlighted a novel aspect of innate immune response directed against MeV unexpectedly involving DNA-related sensing through activation of the cGAS/STING axis, even in the absence of any viral DNA intermediate. In addition, while MeV infection most often causes a mild disease and triggers a lifelong immunity, its tropism for invariant T-cells and memory T and B-cells provokes the elimination of one primary shield and the pre-existing immunity against previously encountered pathogens, known as “immune amnesia”.

## 1. Introduction

Measles virus (MeV), which belongs to the *Paramyxoviridae* family within the order of *Mononegavirales,* is a strictly human pathogen responsible for measles disease. It remains one of the most contagious viruses, with a basic reproduction number R0 comprised between 12 and 18 [1]. The clinical manifestations of measles range from mild (fever, cough, skin rash and conjunctivitis) to severe (pneumonia and encephalitis) [2]. In addition, MeV infection results in a transient immune suppression that increases the risk of opportunistic infections and reduces the efficacy of previous immunizations against other pathogens [1]. While its eradication had been almost achieved during this century in developed countries through systematic vaccination, the increasing doubts towards vaccines in the population, combined with a slow-down in the MeV vaccine campaign due to the COVID-19 epidemics, has led to a large increase in cases, associated with more than 200,000 deaths per year in the last decade [3,4]. Indeed, the lack of adequate vaccination coverage provoked an important decrease in global protection, favoring a context of re-emergence for the virus.

MeV targets lymphocytes and epithelial cells, and can penetrate into the central nervous system via unknown pathways [5]. Wild-type MeV strains bind signaling lymphocytic activation molecule 1 (SLAMF1 or CD150), a lymphoid surface receptor involved in immune cell activation [6]. Following an initial infection of immune cells, MeV is transmitted to epithelial cells by using Nectin-4 receptors localized at their basolateral site [7]. On the other hand, vaccine and laboratory strains use in addition the ubiquitously expressed CD46 receptor to enter cells and replicate [8,9]. One of the main characteristics associated with MeV infection is the depletion of lymphocytes, accompanied by an increase in viremia. Indeed, all subsets of activated lymphocytes, from innate invariant T-cells to memory lymphocyte populations, decrease in number following their direct infection [10,11,12,13]. While MeV infection causes a mild disease and provides life-long immunity in immuno-competent hosts, it dismantles the host’s defenses by infecting and eliminating pre-existing memory B- and memory T-cell subsets and seemingly invariant lymphocyte sub-populations. This process leads to a transient “immune amnesia” and favors opportunistic microbial infections [14,15].

The MeV genome codes for six structural proteins comprising two surface glycoproteins, hemagglutinin (H) and fusion protein (F), along with four internal proteins: matrix (M), nucleoprotein (N), phosphoprotein (P) and large polymerase (L) [16]. Moreover, two non-structural proteins, V and C, are encoded by the *P* gene and mainly act as immunomodulators. MeV can harness both the production of type-I interferons (IFN-I) and the subsequent activation of the IFN stimulated genes (ISGs) [17]. Remarkably, only eight viral proteins confer an arsenal of countermeasures capable of strongly reducing the interferon cascade at different levels, thus helping MeV dissemination.

However, host defenses remain effective in sensing nucleic acids and generating anti-MeV cellular immune responses to later control and clear the infection in the vast majority of cases. MeV displays pathogen-associated molecular patterns (PAMPs), recognized by pattern recognition receptors (PRRs), capable of triggering the innate immune response. Cytoplasmic PRRs, Toll-like receptors (TLRs) and IFN-inducible antiviral molecules recognize nucleic acid-based viral structures and elicit cellular defenses through the expression of IFN-I and consecutive ISGs to implement an antiviral environment [18]. Furthermore, distinct adaptive immune mechanisms are involved in the protection and the final clearance of MeV [19]. Indeed, both T- and B-cell responses against MeV are induced by dendritic cells, leading to helper T-cell stimulation and the activation of cytotoxic CD8^+^ and inflammatory CD4^+^ T-cells, as well as increased antibody production, to achieve viral elimination and the generation of a specific lifelong anti-MeV immunity [10].

While anti-MeV RNA-related innate sensing has been well deciphered, recent studies unveiled that DNA-related sensors are also engaged in the control of paramyxoviruses such as MeV, even though this viral family carries a single-strand RNA (ssRNA) genome without any DNA replication intermediate [20]. Indeed, MeV infection activates the stimulator of IFN genes (STING) through both phosphorylation and ubiquitination, and MeV expansion is favored in the absence of cyclic guanosine monophosphate-adenosine monophosphate (GMP-AMP) synthase (cGAS) and/or STING, thus demonstrating the cGAS/STING axis’ crucial role and opening a new window for further investigations.

## 2. Innate Immunity and MeV Infection

### 2.1. Innate Immune Response against MeV

Following MeV infection, the first line of defense is represented by the IFN response, mediated by the cooperative action of type-I and type-III IFNs [21,22]. This cell-intrinsic immune system triggers the autocrine and paracrine activation of Janus kinase JAK/STAT signaling pathways and the downstream expression of over 300 ISGs, bearing three main functions: (a) the protection of uninfected bystander cells through the establishment of an antiviral state; (b) the priming of innate and adaptive immune responses; and (c) a direct antiviral activity through the autocrine action of virus inhibitory cytokines, such as viperin, which restricts MeV particle release in vitro and in vivo [23].

MeV infection-associated viral PAMPs, such as viral proteins or infection-derived genetic material, are recognized by host PRR, leading to an increase of IFN-I production [24]. The H protein of wild-type MeV strains interacts with TLR2, inducing IL-6 production and the upregulation of MeV entry receptor CD150 expression, thus increasing the number of host cells for efficient viral spreading [25]. The major source of PRR cognate agonists is the viral RNA, transcribed by the viral RNA-dependent RNA polymerase (RdRp) following viral entry into the cytoplasm [18].

### 2.2. RNA-Dependent IFN Response

Cytoplasmic RIG-I-like receptors (RLRs), namely retinoic acid-inducible gene I (RIG-I), Melanoma differentiation-associated protein 5 (MDA-5) and Laboratory of Genetics and Physiology 2 protein (LGP2) are PRRs which are mostly involved in the recognition of MeV RNA in infected cells, thanks to their ability to recognize viral double strand RNA (dsRNA) with or without 5′-triphosphate (Figure 1a) [18,22]. Even though the genome of paramyxoviruses is constituted by (-) ssRNA, dsRNA agonists can be represented by ssRNA with intramolecular dsRNA secondary structures or annealed with other complementary ssRNA [18]. The source of this dsRNA in MeV infection is not yet clear. While it is not likely that (anti)genomic RNA is detected due to its tight packaging into a continuous homopolymer of the viral N protein, small non-coding viral transcripts (leader RNA and trailer RNA) and defective interfering (DI)-derived-RNAs, which are potent inducers of IFN activation, could be possible agonists for RLR activation following MeV infection [18]. Indeed, there is growing evidence suggesting a crucial role of RNA from “defective interferant” subgenomic particles (DI) in the activation of RLRs, as well as from the cellular stress response, as described below [26,27,28,29].

Upon binding to agonist RNAs, while LGP2 acts as regulator of the other RLRs, RIG I and MDA5 recruit mitochondrial antiviral signaling protein (MAVS) through their caspase activation and recruitment (CARD) domains [26,30]. MAVS activates on the one hand interferon regulatory factor 3 and 7 (IRF3, IRF7) through TANK-binding kinase 1 (TBK1) and IkB kinase ε (IKKε)-mediated phosphorylation, and on the other hand the nuclear factor kappa-light-chain-enhancer of activated B cells (NF-kB) through the formation of the IKK complex by IKKα, IKKβ and IKKγ [22]. Both IRF3 and NF-κB are transcription factors that, upon activation, shuttle to the nucleus, triggering IFNβ and inflammatory cytokine expression, contributing to the innate immune response to MeV infection, as well as to the priming of adaptive immune cells [22].

In addition to RLRs, TLR3 and 7 participate to the cell-intrinsic response to MeV by detecting endosomal ds- or ssRNA, respectively [31]. TLR3 signals through TIR domain-containing adaptor proteins, inducing IFNβ (TRIF) to activate IRF3 and NF-κB via TBK1 and IKK complex [32]. In parallel, TLR7 activates the myeloid differentiation primary response 88 (MyD88) adaptor molecule which triggers IKKα-dependent IRF7 activation, leading to the preferential induction of IFNα [32]. While IFNβ is produced in both immune and non-immune cells, IFNα is predominantly expressed in immune cells, since TLR7 is only expressed in some subtypes, such as plasmacytoid dendritic cells (pDCs), which constitutively express high amounts of IRF-7 mRNA [22].

### 2.3. Cellular Stress Response

The IFN response to MeV infection is accompanied by an IFN-inducible dsRNA-dependent stress response, mediated by proteins with direct antiviral activity, such as protein kinase RNA-activated (PKR) and 2′-5′-oligoadenylate synthase (OAS) (Figure 1b) [22].

PKR is an eukaryotic initiation factor 2α (eIF2α) kinase which impairs protein translation by inducing ribosome stalling and stress granule (SG) formation in response to both cellular and pathogen-induced stress [33,34,35]. The formation of SG is associated with the accumulation of the Ras GTPase-activating protein-binding protein 1 (G3BP1), that binds both dsRNA and RIG-I, resulting in RIG-I activation and increased IFNβ expression [36]. The involvement of PKR in the protection against MeV infection has been observed in recombinant MeV vaccinal strains deficient in the C accessory protein (MeV-C^KO^), suggesting an antagonizing activity of MeV-C towards PKR [37,38]. In the absence of MeV-C, PKR is activated at 24 h post infection through phosphorylation on its threonine 466 residue, whereas its activation is impaired by C-expressing MeV strains [37]. Moreover, SG formation occurs in a PKR-dependent manner due to the accumulation of copyback DI dsRNA following infection with the MeV-C deficient virus [27,28]. In contrast, dsRNA recognition by PKR is hampered by the cellular RNA adenosine deaminase ADAR1, through both direct PKR inhibition and dsRNA editing mechanisms [39,40,41,42,43]. In addition to SG formation, PKR exerts an antiviral effect by promoting NF-κB, IRF-1 and mitogen activated protein kinase (MAPK) pathway activation [33,44]. In particular, PKR enhances IFNβ expression through NF-κB activation by inhibiting protein translation, thus reducing the levels of inhibitor κB (IκB) protein [45]. After MeV-C^KO^ infection, PKR increases IFNβ production through the activation of p38/activating transcription factor 2 (ATF-2) and c-Jun N-terminal kinase (JNK)/c-Jun mitogen-activated protein kinase (MAPK) signaling axes and NF-κB, all of which participate in the formation of the IFNβ enhanceosome complex [37,45,46,47].

After sensing dsRNA, OAS synthesizes 2′-5′-oligoadenylate, an agonist of ribonuclease L (RNase L), that implements selective mRNA degradation [48]. The involvement of OAS in the innate response to MeV has not yet been clarified. The RNase L inhibitor ATP-binding cassette subfamily E member 1 (ABCE1 or RLI) has been identified as an essential proviral host factor for MeV [49]. Indeed, viral protein synthesis is strongly reduced in ABCE1-depleted cells, while the host’s protein synthesis is moderately affected. Nonetheless, MeV mRNA levels are not altered by ABCE1 depletion, suggesting that the promotion of MeV infection by ABCE1 could be independent of its RNase L inhibitor activity [49].

### 2.4. A New Player in Anti-MeV Innate Immunity: The cGAS/STING Axis

cGAS/STING is the major innate immune signaling axis responsible for the sensing of double-strand DNA (dsDNA) in cytoplasm [50]. Indeed, while dsDNA is contained in cellular organelles at basal state, it can leak into cytosol during microbial infections, cancer and senescence, through the disruption of nuclear or mitochondrial membrane integrity, or by the transfer of genetic material from dead bystander cells [51]. Following the interaction of the DNA sensor cGAS with cytoplasmic dsDNA, cGAS dimerizes and oligomerizes, catalysing the synthesis of the second messenger cGAMP through the cyclization of guanosine triphosphate (GTP) and adenosine triphosphate (ATP) [52,53]. cGAMP directly activates the endoplasmic reticulum (ER)-resident STING, that translocates to Golgi network and perinuclear puncta, triggering the activation of IRF-3 and NF-κB transcription factors, leading to IFN and pro-inflammatory cytokines expression [54]. While the function of cGAS/STING signaling axis in the innate immune response to DNA viruses is well deciphered, increasing evidence demonstrates its significant contribution to the control of various RNA virus infections [20,55,56,57,58,59,60].

The first in vivo description of a potential involvement of the DNA-sensing cGAS/STING axis in the response to a *Paramyxovirus* infection was obtained in the study aiming to understand the control of Nipah virus (NiV) infection in mice. Even in the absence of the TLR/RLR adaptor molecules MyD88, TRIF and MAVS, mice control NiV infection better than IFNAR KO mice that are unable to respond to IFN-I stimulation [61]. In contrast, the ability to survive NiV infection is completely abolished in quadruple MyD88/TRIF/MAVS/STING KO mice, indicating the crucial and non-redundant role of STING signalling in the response to paramyxovirus infections [20]. In addition, human cell lines depleted for the single cGAS or STING elicited a higher viral replication following MeV infection [20]. Finally, the involvement of the STING axis in response to *Paramyxovirus* infections was confirmed by detecting the main molecular signatures of STING activation, phosphorylation and K63-linked ubiquitination, following MeV and NiV infections in both murine and human cells [20]. This is partially conflicting with the observations previously made in vitro of Sendai virus (SeV), another paramyxovirus belonging to *Respirovirus* genus [62]. Whereas STING modestly restricts the replication of SeV, no hallmarks of STING activation are detected following SeV infection [62]. Paradoxically, a proviral effect of STING could be suggested by a slight increase in IFNβ induction, which is observed in the absence of STING [62]. A similar pattern is observed during infections by vesicular stomatitis virus (VSV), which also belongs to the *Mononegavirales* order but is part of *Rabdoviridae* family. In this case, STING antagonizes virus replication by inhibiting translation initiation through a RIG-I dependent mechanism, showing both a non-canonical mechanism of STING activation and an example of cross-talk between RNA- and DNA-dependent immune signalling pathways [62]. Thus, the potential involvement of STING-dependent translation inhibition during paramyxovirus infections could be investigated in future studies.

Despite the evidence of the contribution of cGAS/STING in the response against paramyxovirus infections, the mechanism of activation of this DNA-sensing axis in the context of an ssRNA virus infection remains poorly understood. One explanation is that MeV induces mitochondrial stress, which is characterized by the downregulation of mitochondrial biogenesis, hyperfusion of mitochondria and release of mtDNA in cytosol through a mitofusin 1 (Mfn1)-mediated mechanism [60]. Cytoplasmic mtDNA can then be detected by cGAS, triggering the activation of cGAS/STING signaling and IFNβ expression (Figure 2) [60].

## 3. MeV Evasion from Innate Immunity

### 3.1. Inhibition of Interferon Cascade

#### 3.1.1. Blockade of IFN-I Production

Following infection, cells can react via a broad range of molecular partners able to recognize viral molecular patterns. IRF-3, IRF-7 and NF-κB are well-characterized transcription factors responsible for the induction of IFN-I expression [18,21]. To sustain its presence in its host, MeV establishes immune escape strategies at various steps of the innate signaling pathways to disrupt the IFN cascade and the implementation of an efficient antiviral environment [63]. One major inhibitor of RLR recognition is the simultaneous and complete encapsidation of nascent genomic and antigenomic RNA by the N protein, which prevents the production of naked 5′ppp RNA and the formation of dsRNA, thus shielding away strong recognition by RIG-I, MDA5 and PKR [18,64]. In addition, P, V and C proteins represent the key players fulfilling that mission by interfering with the gene expression dependent on RLRs, PKR and, in the specific case of pDCs’ infection by vaccinal strains, TLR signaling (Figure 3) [17,65].

The MeV V protein interacts with MDA-5 and LGP2, two of the three innate immune sensors of the RLR family, through its carboxy-terminal domain [66,67]. V protein has also been described as interacting with the ubiquitin-ligase tripartite motif-containing protein 25 (TRIM25), thus hampering RIG-I activation and subsequent IFN-I production [68]. The inhibition of RLR activation by the MeV V protein can also be achieved through the blocking of protein phosphatase 1 (PP1)-mediated dephosphorylation of MDA-5 [69]. In parallel, the MeV V protein binds to the NF-κB p65 subunit, hence retaining it in the cytoplasm and preventing NF-κB-dependent gene expression [70]. Finally, in pDCs infected by MeV vaccinal strains, the V protein impedes the phosphorylation of IRF-7 by serving as a decoy substrate to IKKα [65]. As a consequence, V prevents the interaction between IRF-7 and IKKα, inhibiting the subsequent phosphorylation of IRF-7 and its nuclear translocation [65].

Furthermore, the C protein has been described as altering viral replication by diminishing the levels of DI RNA and its subsequent innate immune recognition by RIG-I and MDA-5 [27,71]. Moreover, MeV-C has the capacity to penetrate the nucleus through nuclear pores and inhibit IFN-β, despite its inability to bind or degrade IRF3 as well as impeding its activation or nuclear translocation, suggesting an interaction with unknown nuclear factors [72]. Apart from the direct inhibition of IFN production, the C protein indirectly antagonizes IFNs’ expression by inhibiting the cellular stress response. As described above, MeV-C decreases the amount of copyback DI RNA, thus preventing its recognition by PKR [27]. The inhibition of PKR activation affects IFNs’ production by two mechanisms: first, the impairment of SG formation and GB3P1-dependent RIG-I activation and second, the blockade of PKR downstream signaling involving NF-κB, IRF-1 and MAPK pathways [27,37,38,41,42]. Finally, recent work has demonstrated that MeV C protein interacts with p65-iASPP protein complex, controlling both cell death and innate immunity pathways [73].

#### 3.1.2. Blockade of IFN-I Signaling

Due to common signaling pathways shared between its respective receptors, the disruption of the IFN cascade by MeV is not limited to the production of IFN-I and/or IFN-III, but extends to its biological activities. Indeed, once released from the infected cells, IFN-I favor the implementation of a strong antiviral environment. However, following autocrine and paracrine stimulation ensuing the binding of IFN-I to IFNAR, MeV harnesses its downstream signaling pathway through synergistic interferences by viral proteins [12,22].

First, the MeV V protein forms a complex with Janus kinase 1 (JAK1) and Tyrosine kinase 2 (TYK2), thus inhibiting their phosphorylation and the subsequent activation of the transcription factors Signal Transducer and Activator of Transcription (STATs) [74]. Second, V and C proteins hamper the phosphorylation of JAK1 when forming a complex with IFNAR1-Receptor for activated C kinase 1 (RACK1)-STAT1 [75,76]. Third, the carboxy-terminal domain of the V protein strongly interacts with STAT2, and the common amino-terminal domain of P and V proteins tightly bind to STAT1, thus obstructing the formation of the STAT1/STAT2 dimer and the downstream expression of ISGs [77,78,79,80].

### 3.2. Innate Immune Amnesia

Recently, MeV-mediated immune amnesia of the innate immunity system has been suggested to be associated with the destruction of two innate-like lymphocyte populations. Mucosal associated invariant T (MAIT) cells are a subset of lymphocytes, representing between 1 and 4% of peripheral blood T-cells and up to 10% of airway T-cells [81,82]. The invariant natural killer T-cells (*i*NKT) are a singular T-cell population that lies at the interface between innate and adaptive immunity [83]. These cell populations fight external pathogenic assault by recognizing specific bacterial antigens presented in the major compatibility complex (MHC). MAIT cells recognize vitamin B2 and B9 metabolites exposed to MHC-related 1 (MR1), while iNKT cells detect glycolipid antigens bound to CD1d (Figure 4). Moreover, the invariant T-cells (iT) are able to stimulate other immune effector cells, such as natural killer cells (NK) and cytotoxic T lymphocytes. MAIT cells express high levels of CD150 on their surface and die by apoptosis upon in vitro infection by MeV [84]. However, despite iNKT and MAIT cells displaying similar levels of CD150 expression and MeV infection, MeV-induced apoptosis of iNKT has not yet been demonstrated [84]. This observation suggests that, upon infection by MeV, MAIT cells are lost, thus constituting an additional mechanism favoring development of the immunosuppression and secondary bacterial infections. Future experiments need to be performed in vivo to ascertain the importance of such a mechanism, by addressing the kinetic of MAIT activation, death and reconstitution following MeV infection and the potential involvement of iNKT cells [84].

## 4. Adaptive Immunity and MeV Infection

### 4.1. Disruption of Pre-Existing Humoral and Cellular Immune Responses by MeV

MeV infection starts in the respiratory tract, where immature pulmonary dendritic cells (DCs) and/or alveolar macrophages capture MeV particles and/or get infected. Then they migrate to lymph nodes, where they promote viral spreading thanks to the abundance of target cells in lymphoid tissues. Indeed, the origin of MeV immunosuppression notably resides in its tropism to CD150-expressing memory T- and memory B-cells [14]. After efficient infection by MeV, innate and adaptive CD150^+^ cells undergo apoptosis, inducing transient lymphoid lineage cytopenia [15]. This process is responsible for the erasure of previous immunological memory and the augmented risk of microbial infection, which are responsible for the most of cases of measles-associated death [12]. At the same time, the immune cells which survive to MeV infection contribute to the establishment of a robust adaptive response, leading to the resolution of the infection, as well as the establishment of a MeV-specific lifelong protection [12].

#### Adaptive Immune Amnesia

One of the major immunopathogenic outcomes during MeV infection is the “reset” of pre-existing adaptive immunity against previously encountered pathogens. As the infection of lymphocytes leads to their apoptosis and subsequent lymphopenia along a transient follicular depletion in CD150-expressing T- and B-cells, it leaves host integrity “wide open” for secondary or latent infections (Figure 5) [15,85]. This phenomenon was reproduced in ferrets by Petrova et al., demonstrating that influenza-vaccinated animals subsequently infected by MeV developed severe influenza when challenged by the influenza virus while MeV infection was combated [86]. The depleted lymphoid organs are rapidly replenished with B- and T-cells, including those specific for MeV [15].

In parallel to the direct interaction/infection leading to elimination of lymphocytes by MeV, indirect environmental disturbances targeting the innate immune system can also contribute to lymphocytic deficiencies. Indeed, glycoprotein H attachment to CD150 on the surface of dendritic cells (DC) provokes an inhibition of IL-12 cytokine production necessary in T-cell maturation and proliferation, and DC antigen presentation to T-cells and MeV H was shown to trigger intracellular signaling in CD150-expressing dendritic cells and inhibit immune response [25,87,88]. Overall, the inhibition of IL-12 secretion abolishes the induction of the Th1 cell population and favors Th2 cell maturation with anti-inflammatory properties. TheTh2 subset produces IL-4 and IL-10 cytokines that further inhibit Th1 cytokines, thus synergistically suppressing the activation of macrophages, the proliferation of T-cells and the implementation of any potential Th1 response [12]. As a consequence of IL-10 excess, the presence of regulatory T-cells is promoted, stepping up the suppression of the immune response [89,90]. Finally, the MeV nucleoprotein was shown to interact with the Fcγ receptor on DC and lead to the strong suppression of the inflammatory responses, similar to the nucleoproteins of the other morbilliviruses [17,91,92,93]. Globally, multiple complementary mechanisms could hamper the ability of the infected host to control and clear the virus, contributing to its persistence and favoring secondary infections.

### 4.2. Adaptive Immune Response against MeV

#### 4.2.1. Measles Virus Replication and Immune Clearance

Following infection, immune cells initiate an adaptive immune response by presenting viral antigens in the context of MHC II and MHC I molecules [11]. Infected cells (mostly B-cells, CD4^+^ and CD8^+^ memory T-cells and activated monocytes) then access the bloodstream and propagate the virus to both lymphoid and non-lymphoid organs, such as the spleen, thymus, kidney, lung, liver and conjunctivae, where MeV infects resident immune cells, as well as endothelial and epithelial cells, whose infection allows MeV excretion and epidemic transmission [11,94].

Viral replication and spreading occurs during a clinically silent incubation period of 10–14 days, followed by the appearance of typical measles symptoms, notably fever, rash, cough and conjunctivitis. Viral clearance then follows within two weeks in immunocompetent individuals [12]. However, even after the clearance of virus-infected cells from the bloodstream, viral RNA can persist within lymph nodes for months, as observed in both humans and macaques [95]. This event likely contributes to different features of measles pathogenesis: (a) prolonged immune suppression; (b) lifelong MeV-specific immunity, enhanced by avidity maturation and germinal centers development through a continuous stimulation by viral antigens in lymphoid tissues; (c) and, in rare cases, the development of late and life threatening neurologic diseases, observed up to months or years post-infection [96,97].

MeV infection is marked with a skin rash in the acute phase of the disease, which is accompanied by the immune clearance of MeV-infected cells. Skin keratinocytes are susceptible to MeV, and their infection, in addition to the dissemination of infected lymphoid and myeloid cells in the epidermis, plays an important role in the pathogenesis of MeV rush [94,98]. Nectin 4-dependent infection of keratinocytes could consequently modulate their cytokine production and may result in a beneficial effect on the progression of skin inflammatory diseases, including atopic dermatitis [94,99].

#### 4.2.2. Development of Specific Anti-MeV Humoral and Cellular Responses

In 90% of immunocompetent individuals, the rash onset is concomitant to the rapid production of MeV-specific IgM, which are maintained for about one month after infection [19]. Moreover, MeV-specific IgG are developed 3–4 days later and are then maintained lifelong [19]. Most anti-MeV antibodies are specific to the N protein, but H, F and, to a lower extent, M protein epitopes are recognized as well [19]. While neutralizing IgG1 and IgG3 specific for H or F proteins are predominant during measles’ acute phase, IgG3 tends to decrease during measles’ convalescent phase and long-term humoral memory. In parallel, high levels of IgG1 and IgG4, with strong complement activation ability, are maintained. [19].

In parallel to the development of the humoral immune response, MeV-specific cytotoxic T cells and soluble indicators of T cell activation, including β2 microglobulin, soluble CD8, CD4 and Fas, are detected as early as rash onset, corresponding to an increase in plasma levels of IFNγ and soluble IL-2 receptors [100]. While the acute phase of measles is dominated by an immunostimulatory T helper 1 (Th1) response, characterized by IFNγ and interleukin 2 (IL-2) secretion, during the convalescent phase, cell-mediated immunity is progressively shifted towards a Th2 profile, with the expression of IL-4, IL-5, IL-10 and IL-13 [19]. Th2 polarization may be linked to MeV-induced immune suppression following the initiation of an effective MeV-specific response and, at the same time, to the prolonged stimulation of B cell maturation, providing lifelong immunity [11,19].

MeV-specific neutralizing antibodies play a major role in the prevention of MeV infection. However, their contribution to viral clearance following the primary infection in naïve hosts seems minor compared to that of cell-mediated immunity [101]. Indeed, children with agammaglobulinemia are normally able to clear MeV infection, while children affected by severe cellular immune deficiencies, such as leukemia, are unable to counteract measles disease [102]. Accordingly, MeV infection of rhesus macaques is followed by a higher viremia when CD8^+^ lymphocytes are depleted, while the depletion of B lymphocytes has no impact [103,104].

As a consequence of the immune impairment perpetrated by MeV, the reconstitution of the immune system leads to the enrichment of an immature naïve B cell population and the reduction in B cell diversity [86]. This phenomenon was recently highlighted by the work of Mina et al., where 77 children were analyzed before and after MeV infection for the diversity of their antibody repertoire [105]. While B cell diversity shrinks, the overall number of memory cells a few weeks after MeV infection remains preserved, as MeV specific clones fill the place left vacant by the pre-existing memory B cells (Figure 6). Consequently, strong anti-MeV immunity provides an effective and lifelong protection after initial infection. Collectively, this phenomenon is known as “immune paradox”. Importantly, the MeV vaccine does not induce an impairment of pre-existing immunity, while the two-doses regimen provides a sterilizing immunity in 97% over a several decade period, even in human immunodeficiency virus positive patients [106].

## 5. Conclusions and Perspectives

The long co-evolution between MeV and humans has led to the development of different viral strategies to suppress and overcome host immune barriers, through finely tuned cellular mechanism [107].

RLRs, TLRs and the IFN-inducible stress response play a crucial role in the limitation of pathogen spreading in the first steps of infection, as shown by the life-threatening inoculation of attenuated viral vaccines in congenital STAT2 or IRF-7 human deficiencies [108,109,110]. The viral RNA species that are recognised by RIGI and/or MDA5 during MeV infection are produced during the transcription step and are strongly favoured by defective interferent truncated genomes, called copyback DI RNA [26,27,28,29,111]. However, the precise nature of PRR agonists during MeV infection in vivo remains to be elucidated. The correct characterization of immunostimulatory RNA during MeV infection could lead to the production of recombinant MeV vaccines capable of increased DI-RNAs generation, representing an intrinsic adjuvant for immune system stimulation [29].

Regarding the cellular stress response, PKR has been shown to be both activated and antagonized during MeV infection [37]. Due to its ability to impair viral spreading through several mechanisms, PKR-dependent response could represent an interesting target to boost the innate immune response against MeV. Moreover, translation inhibition by PKR has been shown to be sufficient to trigger an NF-κB response [45]. Translation inhibition can be achieved through inhibitors of protein phosphatase 1-growth arrest and the DNA damage-inducible gene 34 (PP1-GADD34), a complex responsible for the dephosphorylation of eIF2α [112]. GADD34 inhibitors are particularly promising, since GADD34 is a stress-inducible protein; as a consequence, virus-infected cells would be the primary target of the treatment [113]. By contrast, direct agonists of PKR would target all cells, leading to possible detrimental effects. The GADD34 inhibitor Sephin1 has been tested for its antiviral activity against MeV [114]. Sephin1 blocked MeV replication and increased the level of eIF2α phosphorylation after the exposure of cells to PKR agonists, but not after MeV infection, suggesting that some of its antiviral effects could be independent of p-eIF2α. However, Sephin1 failed to work in vivo due to its toxicity coupled to a modest antiviral effect at low doses [114]. Further studies will thus be required to develop GADD34 inhibitors with lower toxicity and higher efficacy or alternative molecules targeting the PKR-related response.

Furthermore, a role for OAS1/RNase L axis in the response against MeV has been proposed, as ABCE1 was shown to be a proviral host factor for MeV [49]. However, the proviral effect of ABCE1 is independent of its inhibitory activity against RNase L. As ABCE1 is involved in ribosome recycling, it would be worth testing whether the requirement of MeV for ABCE1 could depend on its ribosome recycling function and, in that case, whether ABCE1 could be a potential target for antiviral therapy. Finally, RNase L could still be taken into consideration as a factor involved in the control of MeV infection, as intranasal administration of 2′,5′-oligoadenylate (2–5 A) RNase L agonist was shown in vivo to block viral replication of respiratory syncytial virus (RSV), another paramyxovirus [115].

In addition to canonical RNA-sensing pathways, the cGAS/STING axis can enhance IFN production during MeV infection. Regardless of its ability to detect only dsDNA, cGAS/STING has been recently demonstrated to contribute to the protection from MeV infection in vitro and in vivo [20]. While STING activation through phosphorylation and ubiquitination was detected 24–48 h following MeV infection, the mechanisms of activation are still not fully understood. The late activation and the inability of cGAS to react to viral RNA suggest a delayed activation through self-DNA that could gain access to cytoplasm from the nucleus and/or mitochondria due to several viral-induced types of stress. However, the mechanism triggering mitochondrial damage and the general downregulation of mitochondrial biogenesis observed during MeV infection remains unclear. One possible source of cellular stress during MeV infection could be represented by syncytia formation, provoking mechanical, biochemical and molecular alterations that may explain the extensive mitochondrial remodeling and rupture occurring in MeV-infected cells [116,117]. In line with this hypothesis, IFNβ expression is amplified by cell-to-cell fusion and is associated with sustained nuclear localization of IRF-3, which acts on downstream cGAS/STING and RLRs/TLRs. On the contrary, the upregulation of IRF-7, which is selectively activated downstream RLRs and TLRs, is poorly sensitive to the fusion process [24,118]. Finally, no MeV protein antagonizing cGAS/STING has been identified to date, suggesting the potential targeting of cGAS/STING as therapeutic strategy to overcome MeV-induced IFN inhibition. In addition to mtDNA detection, genomic DNA that could originate from cellular genomes, endogenous retroviruses or surrounding dead cells could be taken into account as a possible alternative source of STING activation, which has not been explored yet. Considering the importance of the cGAS/STING axis in the implementation of both innate and acquired immune responses, future studies regarding the interaction between MeV and cGAS/STING-mediated immunity could lead to important progresses in the understanding and treatment of measles infections [119].

Despite the detection of viral RNA by several host factors, MeV actively suppresses the human innate immune response by blocking IFN production and signaling through its non-structural proteins [22]. Surprisingly, the TLR axis is not directly antagonized by MeV, with the exception of IRF7 inhibition in pDCs, which can be only infected by MeV vaccinal strains. Moreover, contrary to MeV, the ability of the highly pathogenic Nipah virus to antagonize the TLR-dependent signaling through its M and W proteins might explain the difference in virulence between these two paramyxoviruses [120,121,122].

One of the most efficient strategies employed by MeV to achieve the elimination of host immunity is the induction of death in specific immune cell types, leading to the erasure of previous immunological memory. This phenomenon, described as “immune amnesia”, was considered to be mainly due to the MeV-induced T- and B-memory cells’ apoptosis [15]. However, MeV has been recently shown to induce amnesia in the innate immune system as well, by targeting iT cells [13,84]. MAIT cell depletion after MeV infection has been only observed in vitro and the overlap of various surface markers between iT and T or NK cells has prevented MAIT recognition in previous studies, but the specific phenotypic and transcriptional profiling of these cell populations could lead to the elucidation of their role in measles-induced immunosuppression in vivo. Moreover, the utilization of the in vivo knock-out murine models may lead to the identification of more precise MAIT-specific factors, which could allow a better dissection of the effect of each iT cells [123,124]. Finally, the use of iT agonists such as α-galactosylceramide, a glycolipid agonist of iNKT, could represent a promising a therapeutic strategy against immune amnesia during MeV infection [125].

The concurrence between a rapid activation of the IFN response and the inhibition of IFNs and ISGs production by MeV is the basis of active viral replication, which occurs in the absence of symptoms during the incubation period of measles [11]. However, once adaptive immunity is enacted, lymphocyte activation itself contributes to the development of symptoms and, at the same time, to the implementation of a strong cellular and humoral response, leading to viral clearance. Since MeV clearance mainly depends on adaptive immune response, dysregulated functions of antigen-presenting cells responsible for suboptimal recruitment of adaptive-associated immune partners represent an important obstacle. For this reason, both the innate and the adaptive arm of immunity can represent good targets for MeV therapy. Thus, together with the safe and effective anti-MeV vaccine, the enhancement of innate responses through diverse therapeutic strategies affecting early MeV target cell types could contribute to the effective treatment of the infection.

## Figures and Tables

**Figure 1 viruses-14-02641-f001:**
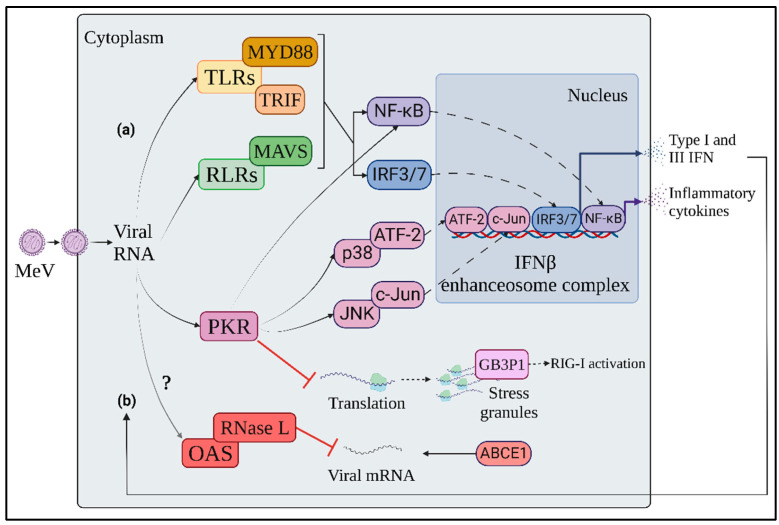
Innate immune response to MeV. (**a**) Soon after MeV penetration into target cells, endo-plasmic or cytosolic viral RNA is detected by TLRs and RLRs, respectively, triggering the activation of MyD88, TRIF and MAVS adaptor molecules. The three axes of RNA-dependent innate immune response converge in the activation of NF-kB, IRF3 and 7 transcription factors, in-ducing both inflammatory cytokines and IFN expression. (**b**) IFN-inducible antiviral molecules, such as PKR and OAS, are activated in an autocrine and paracrine manner through interaction with viral RNA. This IFN-inducible stress response contributes to limiting viral protein production through inhibition of viral translation, selective degradation of viral mRNAs and IFN response enhancement. Orange: TLR signaling axis; Green: RLR signaling axis; Pink: PKR-mediated stress response; Red: OAS-mediated stress response; Blue: IRF transcription factors; Purple: NF-kB transcription factor.

**Figure 2 viruses-14-02641-f002:**
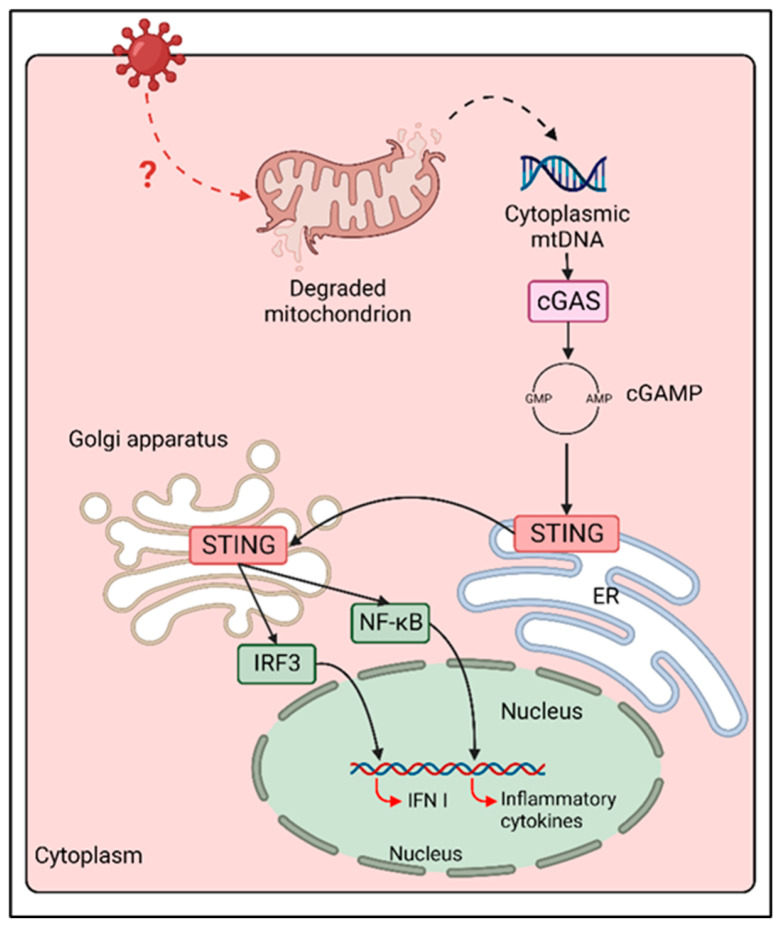
cGAS/STING activation following MeV infection. Following MeV infection, an unknown cellular mechanism induces mitochondrial stress and mitochondrial membrane degradation. mtDNA is than released in cytoplasm, where it is sensed by cGAS, triggering cGAMP synthesis and STING activation. STING translocation from ER to Golgi and perinuclear puncta coincides with the activation of both NF-κB and IRF-3, thus enhancing the expression of inflammatory cytokines and type I IFN during MeV infection.

**Figure 3 viruses-14-02641-f003:**
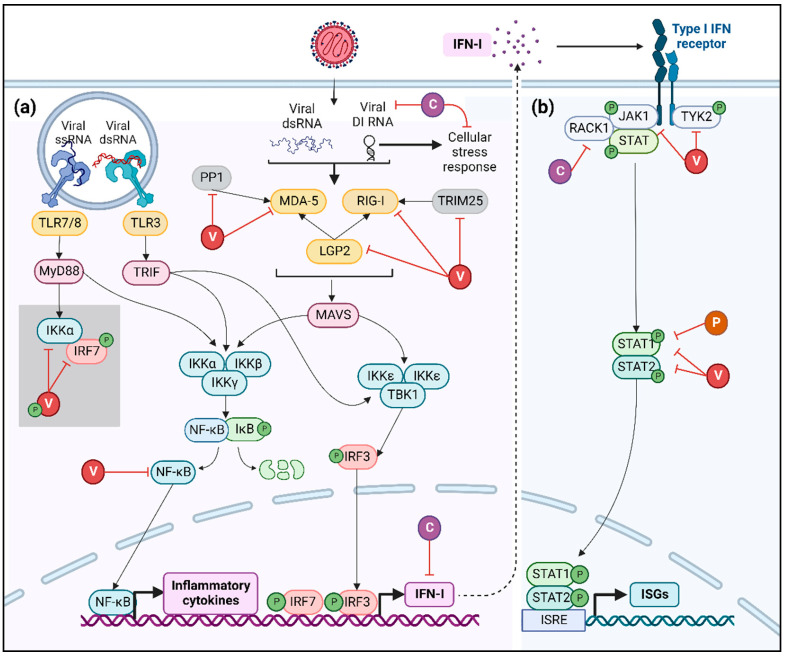
Inhibition of IFN-I cascade. (**a**) Viral RNA recognition by cellular PRR, such as TLRs and RLRs (yellow), activates multiple signaling cascades leading to IFN and NF-κB expression, which is suppressed by V (red) and C (purple) proteins of MeV at diverse levels as summarized here. In the grey square, the mechanism of inhibition of IRF-7 by V protein observed in pDCs infected by MeV vaccinal strains is presented. Phosphate groups are represented in green. (**b**) IFN-I-dependent signaling is inhibited by MeV V, C and P (orange) proteins acting both on JAK/STAT complex and on STAT1/STAT2 transcription factors.

**Figure 4 viruses-14-02641-f004:**
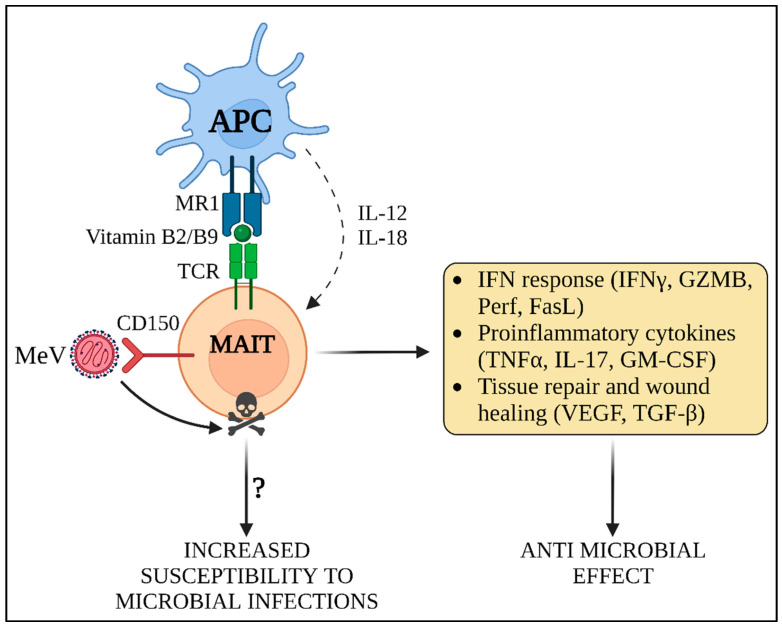
MeV-induced innate immune amnesia. MAIT cells recognize vitamin B2 and B9 metabolites presented in MR1 by antigen presenting cells (APC). In addition, they can be activated independently of MHC antigen presentation through the expression of IL-12 and IL-18 by APCs, which is the predominant mode of activation occurring in viral infections. MAIT cells contribute to the induction of IFN response and cytokine expression, exerting a direct and non-specific anti-microbial activity against invading pathogens. Moreover, they perform additional functions, such as tissue remodeling and wound healing. MeV rapidly infects MAIT and triggers their death by apoptosis. Due to the elimination of MAIT cells, the innate response against both MeV and secondary microbial infections may be suppressed, thus inducing a transient state of “innate immune amnesia”. GZMB: granzyme B; Perf: perforin; FasL: Fas ligand; TNFα: tumor necrosis factor alpha; IL-17: interleukin 17; GM-CSF: granulocyte-macrophage colony stimulatory factor; VEGF: vascular endothelial growth factor; TGF-β: transforming growth factor beta.

**Figure 5 viruses-14-02641-f005:**
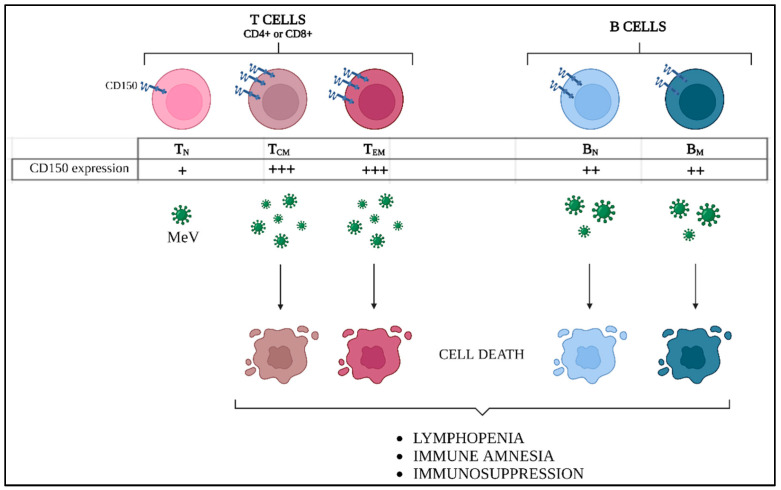
MeV-induced adaptive immune amnesia. Differential CD150 expression levels in lymphocyte populations induce different susceptibility to MeV infection. Central memory (T_CM_) and effector memory (T_EM_) T cells express higher CD150 levels compared to naïve T lymphocytes (T_N_), while intermediate levels are found in both naïve and memory B cells (B_N_ and B_M_). As a consequence, massive cell death of T_CM_, T_EM_, and B_M_ occurs, thus explaining the prolonged lymphopenia and immune suppression during measles disease.

**Figure 6 viruses-14-02641-f006:**
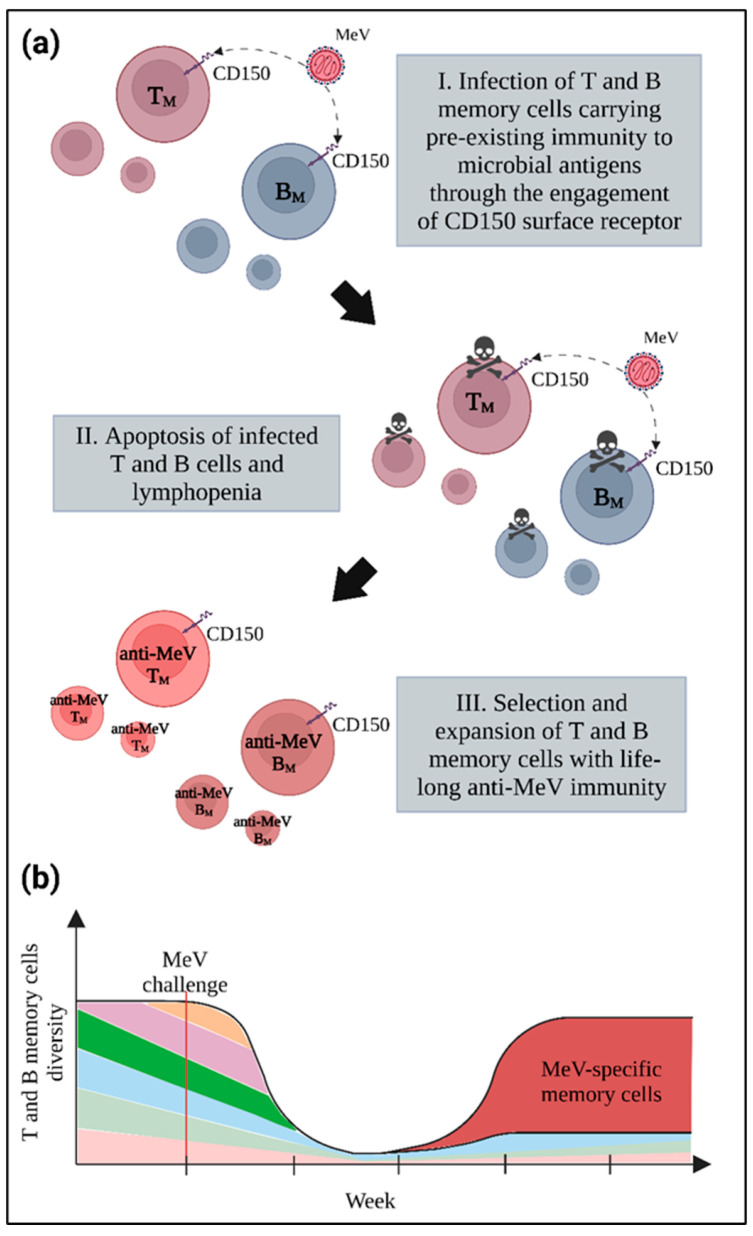
The immune paradox. (**a**) Infection and induction of apoptosis in memory T and B lymphocytes following MeV infection provokes lymphopenia due to the loss of pre-existing immune cells. (**b**) The dramatic shrinkage in memory cell pool diversity towards previously encountered pathogens, represented here by different colors, is associated with the development of a lifelong anti-MeV immunity mediated by the selection and expansion of T and B memory cells with specificity for MeV, represented in red.

## Data Availability

Not applicable.

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
