# Peer review of "Measles Virus-Induced Host Immunity and Mechanisms of Viral Evasion"

_viruses, 2022, doi:10.3390/v14122641_

Round 1

Reviewer 1 Report

In this article Amurri et al. review the current literature on the interplay between Measles virus replication and mechanisms of host immunity. Their work is well structured and properly illustrated by 5 figures. Detailed description of key results and concepts are included as well as discussion of their shortcomings and unresolved questions. Furthermore, perspectives about the relevance of some results for potential therapeutic approaches are also included. The authors also describe and discuss in their work recent discoveries about the role of the DNA sensor cGAS/STING during RNA virus replication.

My recommendation is to accept this review pending minor revisions related to English editing. Below I have listed some minor changes that could be made:

·       12-13: destined in protecting should be destined to protect

·       18: two paradoxical aspects. One could argue that immune recognition of the virus and viral inhibition of immunity are not so much paradoxical as they are confronting or antagonizing aspects of viral infection

·       21: The rRecent work has highlighted…

·     190: …while dsDNA is restricted in cellular organelles… Contained might be more appropriate than restricted.

·     260: NF-kB p65 subunit hence retaining it cytoplasm… Retaining it in the cytoplasm?

·       264: disallowing the subsequent phosphorylation… To disallow means to say officially that something cannot be accepted because it has not been done in the proper way. In the context of the author’s sentence preventing/inhibiting/disrupting might be more appropriate.

·      281-283: The disruption of IFN cascade…their respective receptor. This sentence is not very clear, please revise.

·       310: …the MeV-induced apoptosis of iNKT…

·       526: The ability of the highly pathogenic Nipah virus, differentl from MeV,…

Author Response

In this article Amurri et al. review the current literature on the interplay between Measles virus replication and mechanisms of host immunity. Their work is well structured and properly illustrated by 5 figures. Detailed description of key results and concepts are included as well as discussion of their shortcomings and unresolved questions. Furthermore, perspectives about the relevance of some results for potential therapeutic approaches are also included. The authors also describe and discuss in their work recent discoveries about the role of the DNA sensor cGAS/STING during RNA virus replication.

We thank the reviewer for the good and constructive feedback he provided on the review. We took into account each of his requests and made the specific changes.

My recommendation is to accept this review pending minor revisions related to English editing. Below I have listed some minor changes that could be made:

  • 12-13: destined in protectingshould be destined to protect

We made the correction requested by the reviewer, the text is now: The immune system deploys a complex network of cells and signaling pathways destined to protect host integrity against exogenous threats including Measles virus (MeV).”

  • 18: two paradoxical aspects. One could argue that immune recognition of the virus and viral inhibition of immunity are not so much paradoxical as they are confronting or antagonizing aspects of viral infection

We thank the reviewer for his valuable comment; we changed the text accordingly by: Considerable knowledge has been accumulated regarding MeV interactions with the innate immune system through two antagonistic aspects: recognition of the virus by cellular sensors and viral ability to inhibit the induction of the interferon cascade.”

  • 21: The rRecent work has highlighted…

The text has been corrected and rephrased accordingly by: Recent works have highlighted a novel aspect of innate immune response directed against MeV unexpectedly involving DNA-related sensing through activation of cGAS/STING axis, even in the absence of any viral DNA intermediate.”

  • 190: …while dsDNA is restricted in cellular organelles…Contained might be more appropriate than restricted.

We thank the reviewer for his valuable comment; we whanged the text accordingly by: Indeed, while dsDNA is contained in cellular organelles at basal state, it can leak in cytosol during microbial infections, cancer and senescence, through the disruption of nuclear or mitochondrial membrane integrity or by the transfer of genetic material from dead bystander cells”

  • 260: NF-kB p65 subunit hence retaining it cytoplasm… Retaining it in thecytoplasm?

We thank the reviewer to point that mistake; we corrected the text accordingly by: In parallel, MeV V protein binds to NF-κB p65 subunit hence retaining it in the cytoplasm and preventing NF-κB-dependent gene expression”

  • 264: disallowing the subsequent phosphorylation… To disallow means to say officially that something cannot be accepted because it has not been done in the proper way. In the context of the author’s sentence preventing/inhibiting/disrupting might be more appropriate.

We thank the reviewer for his valuable comment helping us to contextualize rightfully our message, we modified the text accordingly by: As a consequence, V prevents the interaction between IRF-7 and IKKα, inhibiting the subsequent phosphorylation of IRF-7 and its nuclear translocation”

  • 281-283: The disruption of IFN cascade…their respective receptor. This sentence is not very clear, please revise.

We agree with the reviewers that this part of the text was not clear, we modified the text accordingly by: Due to common signaling pathways shared between its respective receptors, the disruption of IFN cascade by MeV is not limited to the production of IFN-I and/or IFN-III but extended to its biological activities. Indeed, once released from the infected cells, IFN-I favor the implementation of a strong antiviral environment. However, following autocrine and paracrine stimulation ensuing binding of IFN-I to IFNAR, MeV harnesses its downstream signaling pathway through synergistic interferences by viral proteins”

  • 310: …the MeV-induced apoptosis of iNKT…

We rephrased that part of the text by: However, despite iNKT and MAIT cells displaying similar levels of CD150 expression and MeV infection, MeV-induced apoptosis of iNKT has not been demonstrated yet”

  • 526: The ability of the highly pathogenic Nipah virus, differentl from MeV,…

We modified the text according by: Moreover, contrary to MeV, the ability of the highly pathogenic Nipah virus to antagonize the TLR-dependent signaling through its M and W proteins might explain the difference in virulence between these two paramyxoviruses”

Reviewer 2 Report

This is an insightful, balanced, and well-referenced review on measles virus interaction with the host immune system. The authors' illustrations are helpful and explanatory. The subject matter is logically laid out – starting with innate immune evasion strategies and ending with the mechanisms underlying the measles immune paradox.   There are several sentences where the English is a bit convoluted, but the review is otherwise written clearly. This review will be a solid addition to the field. I only have very minor comments:

Lines 203-208: While Iampietro & colleagues were likely the first to implicate the cGAS/STING axis in response to paramyxovirus infection in vivo (Ref 61 and 20), I believe the Whelan & Kagan labs (KM Franz et al, PNAS, 2018) showed STING-dependent restriction of multiple RNA viruses including Sendai virus (Genus: Respirovirus Family: Paramyxoviridae). This should be cited and any differences with the authors’ own studies discussed.    

Incomplete reference citation was noted for Ref. 55, 58.

Author Response

This is an insightful, balanced, and well-referenced review on measles virus interaction with the host immune system. The authors' illustrations are helpful and explanatory. The subject matter is logically laid out – starting with innate immune evasion strategies and ending with the mechanisms underlying the measles immune paradox.   There are several sentences where the English is a bit convoluted, but the review is otherwise written clearly. This review will be a solid addition to the field. I only have very minor comments:

We thank the reviewer for the very positive feedback he provided on the review. We replied accordingly to his requests.

Lines 203-208: While Iampietro & colleagues were likely the first to implicate the cGAS/STING axis in response to paramyxovirus infection in vivo (Ref 61 and 20), I believe the Whelan & Kagan labs (KM Franz et al, PNAS, 2018) showed STING-dependent restriction of multiple RNA viruses including Sendai virus (Genus: Respirovirus Family: Paramyxoviridae). This should be cited and any differences with the authors’ own studies discussed.  

We thank the reviewer for his valuable comment and we agree his remark. As requested, we added a paragraph and discussed the work made by Franz et al. accordingly to improve the review. The text has been modified by: “ This is partially conflicting with the observations previously made in vitro on Sendai virus (SeV), another paramyxovirus belonging to Respirovirus genus [62]. Whereas STING modestly restricts the replication of SeV, no hallmarks of STING activation are detected following SeV infection [62]. Paradoxically, a proviral effect of STING could be suggested by a slight increase in IFNβ induction which is observed in absence of STING [62]. A similar pattern is observed during infections by vesicular stomatitis virus (VSV), which belongs also to Mononegavirales order but is part of Rabdoviridae family. In that case, STING antagonizes virus replication by inhibiting translation initiation through a RIG-I dependent mechanism, showing both a non-canonical mechanism of activation of STING and an example of cross-talk between RNA- and DNA-dependent immune signaling pathways [62]. Thus, a potential involvement of STING-dependent translation inhibition during paramyxovirus infections could be investigated in future studies.” 

Incomplete reference citation was noted for Ref. 55, 58.

We thank the reviewer for pointing that discrepancy. Complete reference citations have been added accordingly for both Ref. 55 and 58.